



**Non-Stationary Dynamics of Compound Climate Extremes: A WRF-CMIP6-**
**GAMLSS Framework for Risk Reassessment in Southeastern China**
Yinchi Zhang[1,2,3], Wanling Xu[4], Chao Deng[5], Shao Sun[6], Miaomiao Ma[7], Jianhui Wei[8], Ying Chen[1,
2,3,9], Harald Kunstmann[8], Lu Gao[1,2,3,9*]
*[1]Key Laboratory for Humid Subtropical Eco-geographical Processes of the Ministry of Education,*
*Fujian Normal University, Fuzhou, 350117, China*
*[2]Institute of Geography, Fujian Normal University, Fuzhou, 350117, China*
*[3]School of Geographical Science, Fujian Normal University, Fuzhou, 350117, China*
*[4]School of Ocean and Earth Science, Tongji University, Shanghai, 200092, China*
*[5]School of Geography and Ecotourism, Southwest Forestry University, Kunming 650224, China*
*[6]State Key Laboratory of Severe Weather, Chinese Academy of Meteorological Sciences, Beijing*
*100081, China*
*[7]China Institute of Water Resources and Hydropower Research, Beijing, 100038, China*
*[8]Institute of Meteorology and Climate Research (IMKIFU), Karlsruhe Institute of Technology,*
*Campus Alpin, Garmisch-Partenkirchen, Germany*
*[9]Fujian Provincial Engineering Research Center for Monitoring and Accessing Terrestrial*
*Disasters, Fujian Normal University, Fuzhou, 350117, China*
**\*Corresponding Author: Lu Gao, l.gao@foxmail.com**





**Abstract**
Understanding future changes in compound climate extremes (CCEs) is critical for climate
risk assessment. However, existing research have relied on stationary assumptions, overlooking the
dynamic evolution of CCEs under non-stationary climate change. Therefore, based on an enhanced
Generalized Additive Models for Location, Scale, and Shape (GAMLSS), this study provides novel
perspectives into the non-stationary characteristics of hot-wet (HW), hot-dry (HD), cold-wet (CW),
and cold-dry (CD) extremes under future climate scenarios, focusing on the Minjiang River Basin
(MRB), located in Southeast China. The high-resolution dataset employed for CCEs detection is
generated through dynamical downscaling of a bias-corrected CMIP6 dataset, utilizing the Weather
Research and Forecasting (WRF) model. The results show that (1) CCEs increase significantly at a
rate of 3.55d/10a under the SSP5-8.5 scenario, with hot extremes (HW and HD) playing a dominant
role. The spatial distribution exhibits a distinct west to east increasing gradient, peaking in the MRB
downstream areas. (2) Under the SSP5-8.5 scenario, CCEs exhibit a marked transition from
stationary to non-stationary characteristics, with non-stationarity detected in 95.20% of grid cells.
Mean warming, not variability, served as the dominant factor behind this transition, explaining
80.81% of the changes. (3) The non-stationary results demonstrate that the severity and recurrence
risks of CCEs are systematically underestimated. Most CCEs (except for CD) exhibit increasing
recurrence risks under the SSP5-8.5 scenario, with a trend of 3.12d/10a in the 100-year return period,
showing a stronger increase. This study emphasizes the necessity of updating the risk changes of
CCEs under a non-stationary framework.
**Keywords** compound climate extremes, non-stationarity, GAMLSS, dynamical downscaling, WRF



## 1 Introduction

Global warming is leading to more frequent and intense compound climate extremes (CCEs) (Sauter et al., 2023; Liu et al., 2024; Zhang et al., 2024; You et al., 2025). CCEs have posed severe threats to global social, economic, and ecological systems, with impacts that surpass those of individual extremes in both range and severity (Mukherjee et al., 2023; Zeng et al., 2024; Miao et al., 2024). For example, the Yangtze River Basin in China experienced unprecedented compound hot-dry extremes in August 2022, characterized by record-breaking heatwaves and severe droughts, which directly affected over 50 million people (Jia et al., 2025). The Sixth Intergovernmental Panel on Climate Change (IPCC) report indicated that the probability and intensity of future CCEs are projected to increase (IPCC, 2021). Therefore, a systematic assessment of the future evolution of CCEs is critical for mitigating socio-economic risks and optimizing climate adaptation strategies.

Traditional extreme event analyses rely on stationarity assumptions, presuming that the probability and distributional parameters of climate variables are constant (Sun et al., 2018; Nerantzaki et al., 2023). However, driven by synergistic effects of global warming and anthropogenic forcing, extremes exhibit significant shifts in distributional characteristics (Gao et al., 2018). Therefore, traditional models are not suitable for evaluating extreme changes in the changing environment. To capture these changes, many studies have applied the Generalized Additive Models for Location, Scale, and Shape (GAMLSS) (Rigby and Stasinopoulos 2005) to address non-stationary problems in hydrological and meteorological extremes, enabling updated risk analysis of evolving climate extremes (Lei et al., 2021; Shao et al., 2022; Jin et al., 2023; Li et al., 2024). However, existing non-stationary analyses only focus on individual extremes, and the potential non-stationarity of CCEs has not been established. The comprehensive assessment of future changes in CCEs recurrence risk within a non-stationary framework is also lacking.

The Coupled Model Intercomparison Project Phase 6 (CMIP6) dataset is widely used in climate change research, providing critical predictive understanding of forthcoming climate changes (Singh et al., 2023; Wu et al., 2024; Zhang et al., 2024; Yuan et al., 2024; Feng et al., 2025). While the CMIP6 dataset is applicable for global or large-scale studies, its relatively coarse spatial resolution poses limitations for local-scale investigations (Kim et al., 2020; Abdelmoaty et al., 2021; Zhang et al., 2024). To overcome this constraint, dynamical downscaling, which utilizes nested



high-resolution regional climate models (RCMs), provides a critical technical pathway to
investigate climate response mechanisms at fine-scales (Tapiador et al., 2020; Rahimi et al., 2024).
As an advanced convection-permitting atmospheric modeling system, the WRF model significantly
enhances the simulation capability for meteorological processes at 1-10 km scales through its fully
compressible, non-hydrostatic dynamic core framework (Talbot et al., 2012). Current WRF-based
studies on CCEs predominantly rely on historical reanalysis data, focusing on attribution and
simulation verification of past events (Zhang et al., 2025; Saini and Rohtash, 2025; Deng et al.,
2025). Nevertheless, accurately projecting the evolving trends of future CCEs is crucial for
improving localized disaster resilience and enhancing water security.

In this study, we develop an innovative non-stationary framework for CCE projection through

dynamical downscaling of the bias-corrected CMIP6 (CMIP6bc) data, assessing recurrence risk
evolution during 2025-2065. We focus on the Minjiang River Basin (MRB), a subtropical monsoon-
dominated basin of southeastern China, where complex interactions between topography and
climate give rise to high-intensity compound hydroclimatic extremes. The analysis proceeds as
follows (Fig. 1): Supplement Section S1 presents the validation of CMIP6bc applicability. Section
3.1 characterizes the spatio-temporal patterns of CCEs under both a middle-of-the-road scenario
(SSP2-4.5) and a high-emissions scenario (SSP5-8.5). The non-stationarity detection of CCEs is
described in Section 3.2. The recurrence risk changes in CCEs under non-stationary conditions is
evaluated in Section 3.3. The work establishes a scientific basis for addressing the environmental
and climatic challenges posed by CCEs, thereby contributing to effective strategies for regional
sustainability and climate resilience.



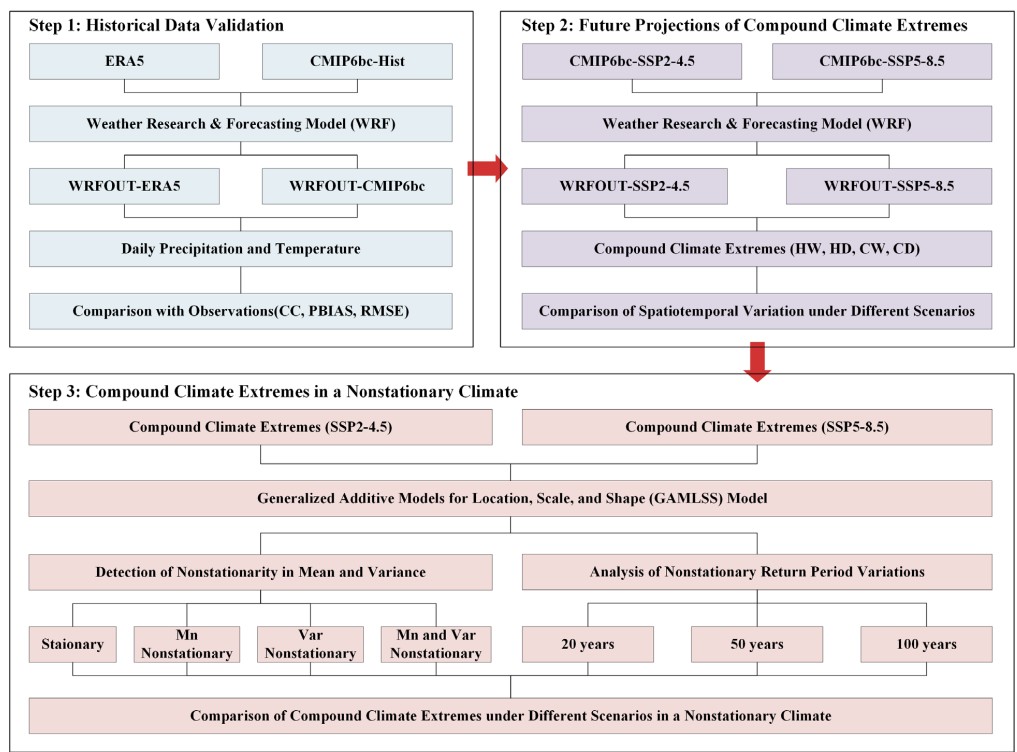


Fig. 1. Flowchart of CCEs projection in a non-stationary framework.



## 2 Study region, methods and data

### 2.1 Study region

The MRB is a complex topographic basin in southeastern coastal China (Fig. 2a). The Minjiang River, the main stream of the basin, drains an area of 60,992 km²–accounting for nearly half Fujian Province's territory. Encompassing three principal tributaries (Jianxi, Futunxi, and Shaxi rivers), the MRB experiences a subtropical monsoon climate characterized by 1700 mm mean annual precipitation and 18°C mean temperature. (Zheng et al., 2023). The basin displays spatio-temporal heterogeneity in precipitation, with flood seasons from April to September that often accompany CCEs.

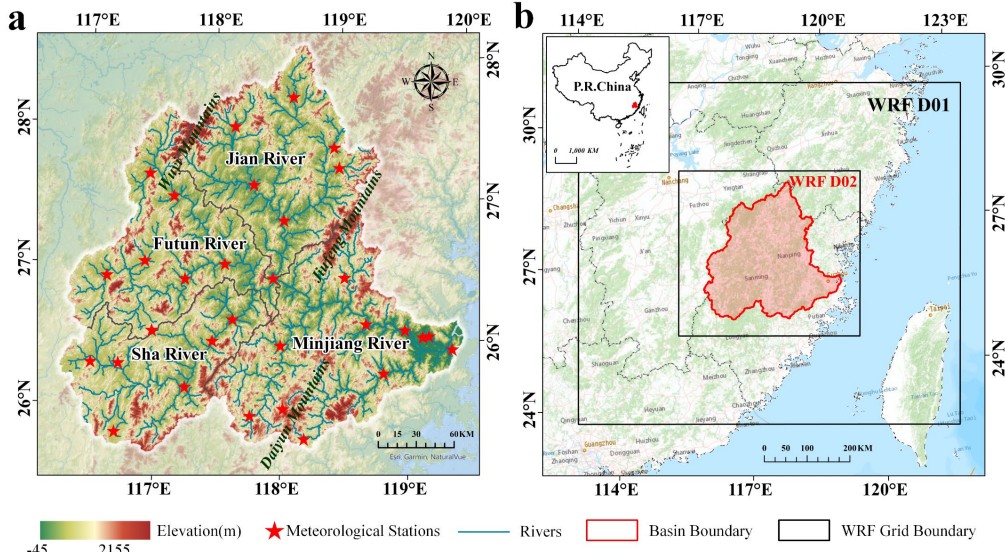

Fig. 2. Study area and model configuration. (a) Topographic features of the MRB (m) and (b) Model configuration with 9-km (D01) and 3-km (D02) nested domains (Zhang et al., 2025). Basemap source: © Esri, https://services.arcgisonline.com





**2.2 Data**

Fujian Provincial Meteorological Bureau provided daily precipitation and temperature records from its 30 monitoring stations. Obtained from the Science Data Bank, the CMIP6bc dataset serves as the foundation for this investigation (Xu et al., 2021, https://www.scidb.cn), which is constructed using the ERA5. This dataset incorporates an 18-model CMIP6 ensemble mean, maintaining both climatological mean and interannual variability statistics while preserving nonlinear temporal trends. Compared with original CMIP6 data, CMIP6bc demonstrates superior performance in extreme event simulation. The dataset used in this study covers the historical period (2005–2014) and future scenarios under SSP2-4.5 and SSP5-8.5 (2025–2065). Moreover, the ERA5, as a widely recognized forcing dataset, was used as a reference to evaluate the performance of historical simulations (Arnault et al., 2021; Jiang et al., 2021; Varga and Breuer, 2022; Shang et al., 2022).

**2.3 Definition of CCEs**

This study considers four types of CCEs: hot-wet events (HW), hot-dry events (HD), cold-wet events (CW) and cold-dry events (CD). We adopt the widely used thresholds (the 90th and 10th percentiles) to identify CCEs (Croitoru et al., 2016; Song et al., 2019; Patel et al., 2024). We first extract daily precipitation (>0.1 mm) and temperature data for each grid during 2025-2065, defining the 90th and 10th percentiles as thresholds to identify hot/cold and wet/dry extremes, respectively. Specifically, we define extreme temperature events as occurring when daily temperatures are higher (hot extremes) or lower (cold extremes) than the threshold. Wet events are characterized by rainfall surpassing the threshold (90th), while dry events are characterized by seven consecutive days without rainfall.



**2.4 Model and experimental design**

2.4.1 WRF model setup

This research utilizes the WRF Version 4.3 with two-domain nested configuration, featuring grid spacings of 9 km and 3 km (Fig. 2b). Table 1 summarizes the optimal physics parameterization schemes selected through our comprehensive sensitivity experiments (Lin et al., 2023; Zhang et al., 2025). At sufficiently high model resolutions, deep convective processes can be explicitly resolved (Arakawa and Jung, 2011). Therefore, the cumulus parameterization scheme is deactivated in the inner domain (D02) to leverage convection-permitting capability. We first simulate daily precipitation and temperature over the MRB from January 1, 2005 to December 31, 2014, using both CMIP6bc and ERA5 forcing data. Subsequently, future projections from January 1, 2025 to December 31, 2065 are conducted using CMIP6bc under two climate projection scenarios.

Table 1 Settings for WRF model in this study.

| WRF model setup overview | | Parameterization scheme settings | |
| --- | --- | --- | --- |
| Forcing data | CMIP6bc, ERA5 | Microphysics | Purdue Lin (Chen and Sun, 2002) |
| Centre | 118.02E°, 26.83N° | Cumulus convection | New Tiedtke (Zhang et al., 2011) |
| Grid | 100×90, 142×130 | Longwave radiation | RRTMG (Mlawer et al., 1997) |
| Resolution | 9km, 3km | Shortwave radiation | Dudhia (Dudhia, 1989) |
| E_vert | 45 | Boundary layer | YSU (Hong et al., 2006) |
| Spin-up time | 7 days | Land surface | Noah-MP (Niu et al., 2011) |

2.4.2 GAMLSS model

GAMLSS is a flexible statistical model used for analyzing distributions with non-stationary characteristics (Rigby and Stasinopoulos, 2005). It extends the traditional generalized linear models (GLMs) and generalized additive models (GAMs) by introducing joint modeling of all distribution parameters (location, scale, and shape). Unlike traditional regression models, GAMLSS effectively characterizes both linear and nonlinear dependencies linking predictors to response variables. (D. M. Stasinopoulos and Rigby, 2007).

This study employs the semi-parametric GAMLSS, which accommodates parametric terms, nonparametric smooth functions, and random effects within a unified modeling structure (Gao et al., 2018). Consider $z$ independent samples $y_i (i = 1, ..., z)$ following a distribution $F_y(y_i \theta_i)$, where the parameter vector $\theta_{iT} = (\theta_{i1}, \theta_{i2}, ..., \theta_{ik})$ contains $k$ components representing location (Mn), scale (Var), and shape (skewness and kurtosis), with $k$ normally not exceeding 4. Model



selection is performed using Akaike's Information Criterion (AIC) (Akaike, 1974), with the optimal
configuration identified through minimum AIC values., and model fitting quality is assessed by the
Filliben correlation coefficient (Filliben, 1975). The GAMLSS is formally defined as follows:
$$g_k(\theta_k) = \emptyset_k \beta_k + \sum_{j=1}^{j_k} h_{j_k}(x_{j_k})$$

where $k$ denotes the indicator of distribution parameters, $\theta_k$ is the distribution parameter vector,
$\emptyset_k$ represents $n \times j_k$ matrix of covariate variables, $\beta_k$ is the coefficient vector of length $j_k$. $g_k(\cdot$
$)$ is the link function connecting distribution parameter to linear predictor. $h_{j_k}(\cdot)$ defines how the
distribution parameter varies with covariate variable $x_{j_k}$. To assess changes in CCE recurrence risk
across the MRB, we fit non-stationary GAMLSS models with two parameters (mean, variance) and
four parameters (mean, variance, skewness, kurtosis) at each grid point, selecting the optimal model
for subsequent analysis. Supplement Table S1 enumerates all distribution functions implemented
in our study. The R code for implementing GAMLSS model can be accessed at
https://github.com/gamlss-dev/gamlss.



## 3 Results

### 3.1 Spatio-temporal patterns of CCEs under future scenarios

Figure 3 (a–j) illustrate the annual spatial distribution characteristics of CCEs in the MRB during 2025–2065. Overall, total CCEs are higher under the SSP5-8.5 scenario (45.48 days) than under the SSP2-4.5 scenario (42.58 days). Specifically, dry extremes (HD and CD), dominate the MRB, while wet-related extremes (HW and CW) occur less frequently. Both hot extremes occur more frequently under SSP5-8.5 than SSP2-4.5, with HW increasing from 0.84 to 1.33 days and HD rising from 22.69 to 27.45 days. In contrast, cold extremes exhibit an opposite trend, with CW decreasing from 3.17 to 2.15 days and CD declining from 15.88 to 14.19 days.

Spatially, both scenarios exhibit similar patterns in CCEs, with the highest frequency occurring in downstream—especially HD and HW, Whereas HD showing a broader distribution, extending to the Futun River and Jian River Basins. Furthermore, CW and CD follow a distinct west-to-east increasing gradient, with highest values near the Jiufeng Mountains.

Temporally (Fig. 3 k–o), the CCEs differ significantly between two emission scenarios. Under high-emission SSP5-8.5, total CCEs increase markedly (3.55d/10a), whereas SSP2-4.5 projects stabilized frequencies. Hot extremes (HW and HD) increase more rapidly under SSP5-8.5, with nearly double rates than those under SSP2-4.5. In contrast, CD shows stronger declining trend under SSP2-4.5 scenario, with approximately 1.7 times than SSP5-8.5.





Fig. 3. Annual Spatio-temporal patterns of CCEs in the MRB from 2025 to 2065.





Having analyzed the annual changes in CCEs, we further identify the seasonal variations of CCEs. As shown in Fig. 4 (a–j), the spatial patterns of CCEs in summer generally align with annual distributions, except for HW, which are rarely observed during this season. Even so, a marked rise in HW under the SSP5-8.5 scenario is evident in the downstream MRB. In summary, total CCEs increase during summer under both scenarios, but with a significantly faster rate under SSP5-8.5 (2.26d/10a and 0.79d/10a). Moreover, the differences in warm events (HW, HD) between scenarios become more pronounced during summer than at the annual scale, while cold events (CW, CD) show consistent patterns (Fig. 4 k–o).

Winter, however, exhibits a contrasting spatial trend. Fig. 5 (a–j) indicate that CCEs primarily occur in the western MRB, particularly concentrated in the Futun River Basin, with higher frequencies under the SSP5-8.5 scenario (11.87 days and 11.51 days). This shift primarily results from the altered spatial distributions of warm extremes (HW and HD), which transition from the downstream MRB to western mountainous areas. Meanwhile, cold extremes (CW and CD) continue to show highest frequency in the Jiufeng Mountain areas. CCEs under SSP5-8.5 also maintain an increasing trend (0.76d/10a), while SSP2-4.5 shows a slight decreasing tendency (Fig. 5 k–o). Among these events, both wet extremes (CW and HW) show insignificant changes, while HD exhibits an increasing trend (1.18d/10a and 0.46d/10a) and CD displays a decreasing tendency (–0.50d/10a and –0.70d/10a).

Given that precipitation and temperature are crucial climate indicators, we calculate average annual values for both over the MRB and investigate their interannual trends. (Supplement Fig. S2). As shown in Fig. S2, precipitation shows slight variation, maintaining relatively stable annual fluctuations. In contrast, temperature demonstrates marked upward progression, particularly accelerated under high-emission SSP5-8.5 conditions (0.46°C/10a). Therefore, we hypothesize that the variation of CCEs in the MRB is primarily controlled by temperature-driven physical processes (with intensifying hot extremes coinciding with declining cold extremes.). Similar findings are also revealed in earlier research (Wu et al., 2020; Zhao et al., 2024; Duan et al., 2024).



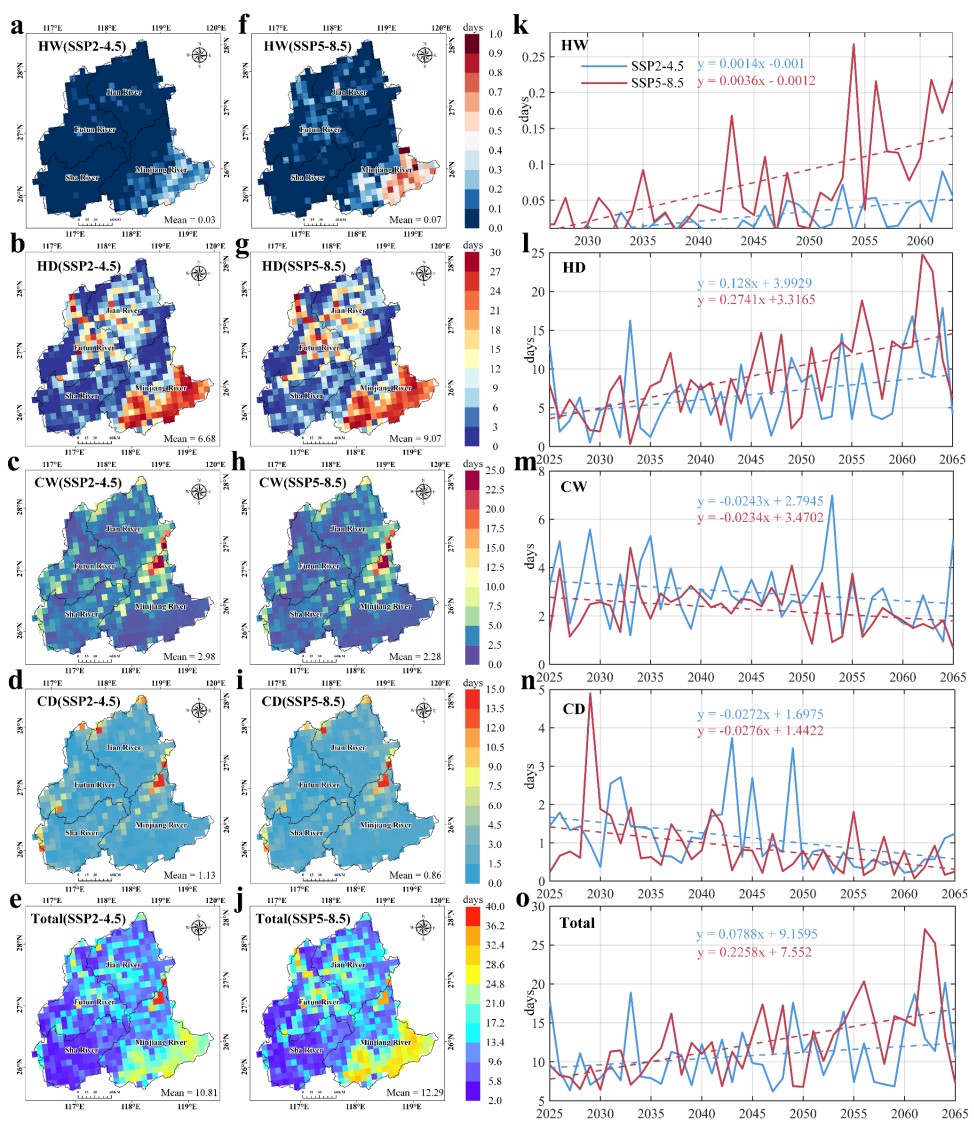

Fig. 4. Same as Fig. 3 but showing results for summer (JJA).



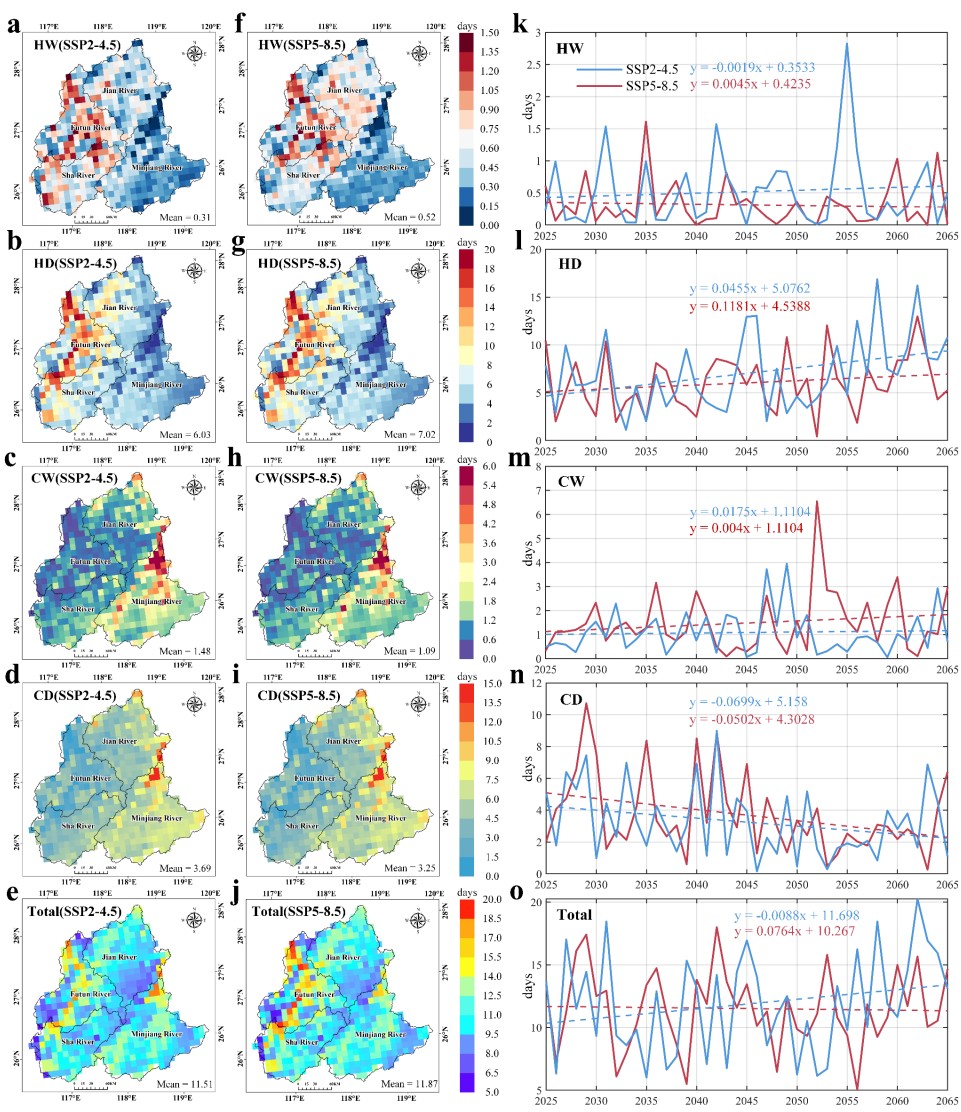

Fig. 5. Same as Fig. 3 but showing results for winter (DJF).





### 3.2 Non-stationary characteristics of CCEs


We examined the variations in both the mean (Mn) and variance (Var) to detect non-stationary
characteristics of CCEs (Fig. 6). The GAMLSS model demonstrated excellent fitting performance
for all indices except HW, as shown by Filliben coefficients exceeding 0.95 in Fig. S3.
Overall, CCEs exhibit a significant transition from stationary to non-stationary characteristics
between SSP2-4.5 and SSP5-8.5 scenarios. This shift is primarily driven by Mn non-stationary
changes, which dominated across 80.81% of grid points in the MRB. Additionally, 11.07% of grid
points are influenced by combined effects of both Mn and Var, primarily distributed in the Shaxi
River Basin and the downstream MRB. Both dry extremes (HD, CD) show a transition from Var to
Mn non-stationarity. Specifically, HD's Mn non-stationarity increasing from 22.65% to 57.03% of
grids and becoming dominant (60%) in downstream areas where stationarity previously prevailed
(54.71%). The Mn non-stationarity of CD also almost covered the entire downstream MRB. For
wet extremes (HW and CW), Mn and Var non-stationarity shows a notable increase, with HW
expanding from 34.30% to 54.02%, and CW experiencing a stronger rise from 0.06% to 23.25%.
In summary, CCEs under SSP5-8.5 demonstrate more pronounced non-stationary characteristics,
with dry extremes primarily driven by Mn changes and wet extremes influenced by combined Mn
and Var effects.
Figure 7 further illustrates the variations in Mn and Var of CCEs. It is clear that Mn exhibits
more pronounced variations compared to Var. For warm extremes, Mn exhibits significant increases
across the entire basin under both SSP2-4.5 and SSP5-8.5 scenarios (at the 99% confidence level),
indicating that climate warming predominantly amplifies the mean frequency of compound heat
extremes rather than their temporal variability. For cold extremes, CD exhibits more pronounced
variations compared to CW, and these changes remain predominantly driven by the reduction in
Mn. Overall, under the SSP5-8.5 scenario, Mn of CCEs shows a significant increase across nearly
the entire basin, while under the SSP2-4.5 scenario, it remains relatively stable. In contrast, Var
exhibits only slight changes under both scenarios.





Stationary    Mn non-stationary    Var non-stationary    Mn and Var non-stationary
Fig. 6. Stationary and non-stationary characteristics for CCEs in the MRB (a-e and k-o), percentage
of non-stationary and stationary characteristics across five basins (f-j and p-t).

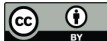

Fig. 7. Results of Mann-Kendall test for Mn (a-j) and Var (k-t), showing the spatial distribution of
Z values. Z values indicate trend significance: |Z| > 1.65 denotes 90% confidence, while |Z| > 2.58
corresponds to 99% confidence.



### 3.3 Changes in the recurrence risks of CCEs

We fit both a non-stationary model and stationary models to investigate the changes in recurrence risk of CCEs. Figure 8 (a-j) presents substantial differences in return period of CCEs under stationary and non-stationary conditions. Our results reveal that stationary models significantly underestimate the recurrence risks of CCEs in future decades, whereas non-stationary models better capture their evolving characteristics. These discrepancies between the two models become more significant over time.

Under the SSP5-8.5 scenario, non-stationary projections show significant increases in the 20-, 50-, and 100-year CCEs. Total CCEs are projected to increase at a rate of 3.12d/10a in 100-year return period. Notably, the stationary models systematically underestimate the risks of CCEs after 2045. Specifically, the recurrence risks of HW (0.36d/10a and 0.15d/10a), HD (11.65d/10a and 2.53d/10a), and CW (0.30d/10a and 0.06d/10a) all show increasing trends, particularly under the SSP5-8.5 scenario. In contrast, CD exhibits a decreasing trend, with a more rapid decline under the SSP2-4.5 scenario (-3.38d/10a). The 100-year CCEs demonstrate higher sensitivity to climate change, highlighting amplified non-stationary effects on these high-impact extremes.

We further quantify the deterministic trends of recurrence risks in CCEs based on the Empirical Mode Decomposition (EMD, Supplement Method S1). Figure 8 k-t present the variations in 100-year recurrence risks, while the complete results for all return periods are provided in Supplement Fig. S4. Overall, CCEs show strong upward trends under the SSP5-8.5 scenario, along with significant spatial heterogeneity. The frequency of CCEs generally shows a decreasing trend from west to east, featuring concentrated high-risk regions in the Shaxi River Basin (some grid points exceed 12d/10a). While all indices except CD show increasing trends, HD emerges as the most severe, with basin-wide increases exceeding 8d/10a. In contrast, CD shows decreasing trends in both scenarios, with a less pronounced decline under SSP5-8.5, suggesting that greenhouse warming has not substantially mitigated recurrence risks associated with CD.

Fig. 8. Comparison of non-stationary (NS) and stationary (S) characteristics for CCEs under 20-, 50-, and 100- year return periods (a-j). Spatial distributions of trends in CCEs under 100-year return periods (k-t), 20- and 50-year return period result are provided in Supplement Fig. S4.



## 4 Discussion

Although earlier research has highlighted the necessity of analyzing extreme events under non-stationary conditions (Cheng et al, 2014; Byun and Hamlet, 2020; Liu et al., 2024), the evolution of CCEs within a non-stationary climate remains lacking. This study develops an innovative non-stationary framework to assess future recurrence risk changes in CCEs, combining WRF with advanced GAMLSS. Our analysis suggests that traditional stationary models may underestimate CCE frequencies. Therefore, updating risk assessments in time under non-stationary conditions is essential to avoid misleading projections and support more robust climate adaptation strategies (Abdelmoaty and Papalexiou, 2023). This innovative framework enables regional-scale reassessment of CCEs which is transferable elsewhere.

The projected increase in CCEs is consistent with global trends of intensifying hydroclimatic risks under continued warming (Asadieh and Krakauer, 2017; Zhang et al., 2021; Shu et al., 2024). Yin et al (2025) indicate that hot-stagnation and hot-dry extremes as the most prevalent CCE types in Eastern Asia, suggesting that temperature variations predominantly influence the occurrence of CCEs in this region. Increasing empirical evidence (Li et al., 2021; Li et al., 2023; Engdaw et al., 2023) reveals that sustained global warming is associated with a rising frequency of hot extremes and a systematic decline in cold extremes. This reversal has been linked to enhanced radiative forcing from anthropogenic greenhouse gas emissions (Samset et al., 2018; Kramer et al., 2021). Our study reveals that future variations in CCEs are predominantly driven by climate warming-induced mean-state shifts rather than enhanced variability. This aligns with global-scale findings that thermodynamic effects (e.g., rising baseline intensity of extremes due to warming) dominate mean-state changes (Horton et al., 2016; Van Der Wiel and Karin, 2021; Nordling et al., 2025). Moreover, conventional stationary models, which rely on fixed statistical assumptions, may fail to capture the escalating severity of future extreme events (Feng et al., 2020; Xu et al., 2025). Our results empirically validate that non-stationary frameworks provide significantly improved estimates of recurrence risk shifts in compound extremes compared to stationary models.

To refine coarse-resolution global climate models (GCMs) outputs, two prevalent downscaling strategies have been established: dynamic and statistical techniques. (Sachindra et al., 2018; Xu et al., 2019). Compared with traditional statistical downscaling approaches, the dynamical





downscaling framework offers significant advantages in representing the physical mechanisms
(Gutmann et al., 2011; Guyennon et al., 2013). WRF model is adept at explicitly resolving
atmospheric dynamics, surface processes, and land-atmosphere feedback mechanisms (Powers et
al., 2017). This capability is especially crucial in the MRB, where complex terrain and significant
surface heterogeneity prevail. Furthermore, extensive research confirms that simulation fidelity
fundamentally depends on initial and boundary condition quality. (Comin et al., 2018; Gholami et
al., 2021; Bello-Millá et al., 2024). Therefore, instead of relying on traditional ensemble prediction,
we use bias-corrected CMIP6 dataset, addressing some of the uncertainties at their source. This
dataset has also been validated for its reliability (Zhang et al., 2024; Yang et al., 2025; Duan et al.,

2025).

Several limitations merit consideration in this study. Firstly, although dynamical downscaling

with the WRF model improves spatial resolution, systematic biases remain in precipitation
simulations over complex terrain. Secondly, the current GAMLSS framework only considers time
as a covariate. Future studies could integrate machine learning approaches for WRF output post-
processing (Yin et al., 2021; Xie et al., 2023) while simultaneously incorporating physical
covariates (e.g., climate drift, circulation indices) to enhance dynamical modeling frameworks
(Zeng et al., 2024; Ma et al., 2025).
**5 Conclusions**

Through this intensive case analysis, we establish a transferable framework for assessing the

non-stationarity of CCEs. This work advances the understanding of the evolution of CCE
recurrence risks under climate change and offers important perspectives to support adaptive
strategies and strengthen disaster risk governance. This study reveals the following important
findings.
1) CCEs increase significantly across the MRB, with trends under SSP5-8.5 (3.55d/10a) scenarios

surpassing those under SSP2-4.5 scenarios. HD extremes dominate spatially (downstream-

focused) and seasonally (summer-peaked), rising at 2.26d/10a, whereas cold extremes decline.

These shifts are primarily temperature-driven, as pronounced warming amplifies hot extremes

but suppresses cold extremes.

2) CCEs exhibit a pronounced shift toward Mn-dominated non-stationarity under SSP5-8.5



scenarios, contrasting sharply with the stationarity in SSP2-4.5. Spatial analysis reveals that 80.8%
of the MRB is governed by Mn-driven non-stationarity under SSP5-8.5, with dry extremes (HD,
CD) showing the most abrupt transitions. For HD, Mn non-stationarity expands from 22.7% to
57.0% of the basin, dominating 60% of downstream grids, showing an increase of nearly three
times compared to SSP2-4.5. CD's Mn-driven shifts cover >90% of the downstream MRB. Var
contributes minimally across both scenarios, confirming that warming amplifies extremes
primarily through baseline intensity shifts rather than stochastic fluctuations.
3) Non-stationary modeling reveals systematic underestimation of CCEs recurrence risks by
stationary approaches. Under SSP5-8.5 scenarios, most CCEs (except CD) exhibit increasing
recurrence risks. Climate change impacts are significantly amplified for 100-year CCEs
(3.12d/10a), as evidenced by their heightened non-stationary responses. Spatial analysis reveals
a distinct east-to-west gradient in recurrence risk, with significantly elevated risk observed in the
western mountainous areas.
**Acknowledgements**
The 'High Performance Computing Center' at Fujian Normal University provided
computational resources for the WRF model simulations.
**Financial support**
Supported by the National Natural Science Foundation of China (Grant No. 42271030), Fujian
Provincial Funds for Distinguished Young Scientists (Grant No. 2022J06018), the Scientific Project
of Fujian Provincial Department of Science and Technology (Grant no. 2022Y0007), the German
Federal Ministry of Education and Research (BMBF) through funding of the KARE_II project
(01LR2006D1) and the 'Young Eagle Plan' Top Talents of Fujian Province.
**Code/Data availability**
Code/Data will be made available on request.
**Declaration of competing interest**
The authors declare that they have no known competing financial interests or personal
relationships that could have appeared to influence the work reported in this paper.



**Author contribution**
Conceptualization: YZ. Methodology: YZ, LG, WX, CD, MM, JW, HK. Software: YZ, WX,
CD. Data curation: SS. Writing- Original draft preparation: YZ. Writing- Reviewing and Editing:
WX, LG. Supervision: SS, MM, YC, HK. Funding acquisition: LG, JW, YC.

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
