# Peer review of "Non-Stationary Dynamics of Compound Climate Extremes: A WRF-CMIP6- GAMLSS Framework for Risk Reassessment in Southeastern China"

_EGUsphere, 2025_

## Author Comment (AC1)

**RESPONSE TO REFEREES**

**Ms. Ref. No.:** Egusphere-2025-2438

**Title:** Non-Stationary Dynamics of Compound Climate Extremes: A WRF-CMIP6-GAMLSS Framework for Risk Reassessment in Southeastern China

**To Anonymous Referee #1**

Thank you for your thoughtful and detailed review of our manuscript. We greatly appreciate the time and effort you dedicated to providing feedback. Your constructive suggestions have been invaluable, and we will implement all changes in a revised version of the manuscript.

1.In the introduction, where you mention the need for fine-resolution models, you may need to add the reasons for the required high-resolution models for capturing compound extremes (e.g., local convective precipitation, spatial heterogeneity, …).

We thank the reviewer for this valuable comment. Following your suggestion, we will add in the revised manuscript the reasons why high-resolution models are necessary to capture compound climate extremes (CCEs), emphasizing factors such as local convective precipitation and spatial heterogeneity.

Compared to Global Climate Models (GCMs), Regional Climate Models (RCMs) offer higher spatial resolution, allowing for more precise simulations of local climate effects induced by topography, such as local convective precipitation, orographic effects, and regional climate heterogeneity (Gilbert et al., 2025). In regions with complex terrain, RCMs are particularly effective at capturing spatial variations of climate variables, such as the differences in wind patterns, precipitation, and their distribution caused by topography in mountainous or basin areas (Imran and Evans, 2025). For example, Byun et al. (2023) assessed the ability of RCMs and GCMs to simulate storm tracks in East Asia, revealing that RCMs are better able to capture high-resolution topography, thereby reducing the biases found in GCMs. Lin et al. (2022) showed that RCMs driven by ERA-Interim reanalysis data are capable of capturing small-scale processes, such as orographic effects, and outperform GCMs in reproducing the large-scale features of the Heat Wave Magnitude Index-daily (HWMId). Torrez-Rodriguez et al. (2023)

also demonstrated that RCMs are better at reproducing the main spatio-temporal characteristics of precipitation in subtropical complex terrain regions.

2.May you please check the following articles and discuss how you improve their work and what your innovation is compared to it? https://link.springer.com/article/10.1007/s00382-020-05538-2

Thank you for your insightful comment regarding the comparison of our work with the study by Singh et al. (2021) on non-stationary compound extreme events (CCEs). After reviewing their work alongside our own, we would like to point out the following key differences and innovations that distinguish our study. Additionally, we will include this comparison in the discussion section of the revised manuscript.:

Compared to the study by Singh et al. (2021), our research presents significant innovations in both spatial scale and methodology. First, while Singh et al. (2021) focus on large-scale ensemble simulations to analyze compound extreme events in Canada, our study targets the medium-to-small scale of the Minjiang River Basin (MRB), a region heavily influenced by complex monsoonal climates and topography. Through high-resolution WRF model dynamic downscaling, we are able to conduct a more fine-grained risk assessment of compound extreme events and their spatial heterogeneity, offering a more localized approach to climate risk evaluation. Second, in terms of methodology, Singh et al. (2021) employ a Bayesian Copula model to analyze the dependence structure between temperature and precipitation, whereas we introduce the GAMLSS model to capture the non-stationary changes and assess the risk of compound extremes. Through the GAMLSS model, we are able to simultaneously handle variations in both the mean and variance of climate variables, providing a more comprehensive and detailed framework for risk assessment.

3.May you make a list of the 18 climate models you applied for your study in the supplementary file?

Thank you for your suggestion. We will provide a list of the 18 models used in the bias-corrected CMIP6 dataset (Xu et al., 2021) in the supplementary file as requested.

Table S1 CMIP6 models used in CMIP6bc.

| No. | Model | Institution | Approximate grid spacing |
|---|---|---|---|
| 1 | ACCESS-CM2 | Commonwealth Scientific and Industrial Research Organisation (Australia) | 1.875° × 1.25° |
| 2 | ACCESS-ESM1–5 | Commonwealth Scientific and Industrial Research Organisation (Australia) | 1.875° × 1.25° |
| 3 | CanESM5 | Canadian Centre for Climate Modelling and Analysis (Canada) | 2.81° × 2.81° |
| 4 | BCC-CSM2-MR | Beijing Climate Center (China) | 1.125° × 1.125° |
| 5 | FGOALS-f3-L | Institute of Atmospheric Physics, Chinese Academy of Sciences (China) | 1.25° × 1° |
| 6 | FGOALS-g3 | Institute of Atmospheric Physics, Chinese Academy of Sciences (China) | 2° × 2.25° |
| 7 | EC-Earth3 | European EC-Earth Consortium (Europe) | 0.70° × 0.70° |
| 8 | EC-Earth3-Veg | European EC-Earth Consortium (Europe) | 0.70° × 0.70° |
| 9 | IPSL-CM6A-LR | Institute Pierre Simon Laplace (France) | 2.5° × 1.26° |
| 10 | AWI-CM-1-1-MR | Alfred Wegener Institute, Helmholtz Centre for Polar and Marine Research (Germany) | 0.94° × 0.94° |
| 11 | MPI-ESM1-2-HR | Max Planck Institute for Meteorology (Germany) | 0.94° × 0.94° |
| 12 | MPI-ESM1-2-LR | Max Planck Institute for Meteorology (Germany) | 1.875° × 1.875° |
| 13 | MIROC6 | Japan Agency for Marine-Earth Science and Technology (Japan) | 1.41° × 1.41° |
| 14 | MRI-ESM2-0 | Meteorological Research Institute, Japan Meteorological Agency (Japan) | 1.125° × 1.125° |
| 15 | NorESM2-LM | Norwegian Climate Center (Norway) | 2.5° × 1.875° |
| 16 | CESM2 | Climate and Global Dynamics Laboratory, National Center for Atmospheric Research (USA) | 1.25° × 0.94° |
| 17 | CESM2-WACCM | Climate and Global Dynamics Laboratory, National Center for Atmospheric Research (USA) | 1.25° × 0.94° |
| 18 | GFDL-ESM4 | Geophysical Fluid Dynamics Laboratory, National Oceanic and Atmosphere Administration (USA) | 1.25° × 1.0° |

4.You mentioned that "The dataset used in this study covers the historical period (2005–2014)", why did you choose only 10 years as historical data?

Thank you very much for your insightful comment. We would like to clarify that the bias-corrected CMIP6 dataset (Xu et al., 2021) we used has already been extensively validated (Jamal et al., 2023; Huang et al., 2024; Wu and Zheng, 2023). Given that the focus of our study is on assessing the non-stationary changes of future compound climate extremes (CCEs), a comprehensive and detailed evaluation of the dataset was not conducted. Additionally, running WRF simulations requires substantial computational resources—for example, simulating one year over the MRB takes approximately 4 days on 80 CPU cores. Considering both the reliability of the dataset and the need to optimize computational resources, we select a 10-year historical period (2005–2014) as being sufficient to demonstrate the reliability of the bias-corrected data for our study purposes.

Thank you for your insightful comment. The reason we focused on SSP2-4.5 and SSP5-8.5 scenarios is that the bias-corrected dataset we used (Xu et al., 2021) only provides data for these two scenarios. This limitation is due to the specific selection of SSP scenarios made during the dataset development. We speculate that the dataset creators aimed to highlight the differences between the moderate and high-emission scenarios, particularly to emphasize the potential impacts of future climate changes under these contrasting pathways. By focusing on SSP2-4.5 (moderate emission scenario) and SSP5-8.5 (high-emission scenario), the dataset offers a clear comparison of how differing levels of greenhouse gas emissions can influence climate projections, especially in terms of temperature rise, extreme events, and other key climate variables. Although SSP1-2.6, which represents a low-emission scenario, is also highly relevant, it may not have been included to maintain the dataset's focus on the more critical scenarios that are likely to dominate future climate projections.

6.Please make clearer how the downscaling integrates with your GAMLSS framework.

We perform dynamical downscaling using the WRF model to downscale the bias-corrected CMIP6 data (1.25° × 1.25°) to a 3 km resolution. Subsequently, compound extreme climate events (CCEs) are calculated based on the high-resolution WRF outputs. These

calculated CCEs are then used as input for the GAMLSS framework to analyze their non-stationary characteristics. This approach allows the integration of high-resolution dynamical downscaling with the statistical modeling of extremes, ensuring that both local-scale variability and non-stationarity are adequately captured. We will provide a more detailed description of this method in the revised manuscript.

7.Please make it clear what you mean by "enhanced" or "advanced" GAMLSS in the manuscript. Do you mean the GAMLSS, which considers non-stationary characteristics?

Thank you for your valuable comment. We apologize for the ambiguity in the description of "enhanced" or "advanced" GAMLSS. We clarify it as follows: Most existing studies apply the two-parameter GAMLSS that only models the mean and variance to constrain data fitting. In contrast, our "enhanced/advanced GAMLSS" extends this framework by incorporating four parameters—mean, variance, skewness, and kurtosis—to comprehensively characterize the data distribution (line 158-160). This extension enables the model to capture more complex distributional features (e.g., asymmetry and tail thickness) that cannot be fully described by the two-parameter setting. We will revise the relevant descriptions in the manuscript to explicitly define this parameter extension, ensuring clarity for readers.

8.In the supplementary file, please provide a table showing the validation results.

Thank you for your comment. We will provide several tables showing the validation results in the supplementary file.

Table S2 Meteorological stations information.

| ID | Name | Longitude (°N) | Latitude (°E) | Elevation (m) |
|----|------|----------------|---------------|---------------|
| 1 | Jiuxianshan | 118.1 | 25.72 | 1653.5 |
| 2 | Gutian | 118.73 | 26.58 | 361.5 |
| 3 | Datian | 117.83 | 25.7 | 400.1 |
| 4 | Youxi | 118.15 | 26.17 | 126.1 |
| 5 | Dehua | 118.23 | 25.48 | 521.4 |
| 6 | Yongtai | 118.93 | 25.87 | 85.6 |
| 7 | Fuzhou | 119.28 | 26.08 | 83.8 |
| 8 | Changle | 119.5 | 25.97 | 4.1 |
| 9 | Minhou | 119.15 | 26.15 | 57.8 |
| 10 | Minqing | 118.85 | 26.23 | 40.8 |
| 11 | Sanming | 117.62 | 26.27 | 215 |
| 12 | Ninghua | 116.63 | 26.23 | 358.9 |
| 13 | Yong'an | 117.35 | 25.97 | 206 |
| 14 | Shaxian | 117.8 | 26.4 | 120.6 |
| 15 | Qingliu | 116.85 | 26.2 | 310.6 |
| 16 | Liancheng | 116.75 | 25.72 | 380.0 |
| 17 | Guangze | 117.3 | 27.52 | 265.4 |
| 18 | Nanping | 118.17 | 26.65 | 125.6 |
| 19 | Jiangle | 117.47 | 26.73 | 154.7 |
| 20 | Jianning | 116.85 | 26.83 | 342.3 |
| 21 | Shaxi | 117.15 | 26.4 | 357.4 |
| 22 | Taining | 117.17 | 26.9 | 342.9 |
| 23 | Shaowu | 117.47 | 27.33 | 191.5 |
| 24 | Shunchang | 117.8 | 26.8 | 175.2 |
| 25 | Jianou | 118.31 | 27.05 | 154.9 |
| 26 | Jianyang | 118.12 | 27.33 | 196.9 |
| 27 | Zhenghe | 118.82 | 27.37 | 221.5 |
| 28 | Songxi | 118.8 | 27.52 | 205.4 |
| 29 | Wuyishan | 118.03 | 27.77 | 220.6 |
| 30 | Pucheng | 118.53 | 27.92 | 276.9 |

Table S3 Temperature simulation results based on ERA5

| Station | STD | RMSE | CC |
|---|---|---|---|
| Jiuxianshan | 1.14 | 0.59 | 0.99 |
| Gutian | 0.98 | 0.29 | 0.99 |
| Datian | 0.97 | 0.07 | 1.00 |
| Youxi | 0.99 | 0.23 | 0.99 |
| Dehua | 0.98 | 0.20 | 0.99 |
| Yongtai | 1.00 | 0.22 | 0.99 |
| Fuzhou | 0.97 | 0.26 | 0.98 |
| Changle | 0.92 | 0.20 | 0.98 |
| Minhou | 0.97 | 0.35 | 0.95 |
| Minqing | 0.98 | 0.16 | 0.99 |
| Sanming | 1.03 | 0.30 | 0.99 |
| Ninghua | 0.99 | 0.12 | 0.99 |
| Yongan | 1.01 | 0.32 | 0.99 |
| Shaxian | 1.03 | 0.36 | 0.99 |
| Qingliu | 0.96 | 0.14 | 0.99 |
| Liancheng | 1.01 | 0.27 | 0.99 |
| Guangze | 1.01 | 0.19 | 0.99 |
| Nanping | 1.02 | 0.29 | 0.99 |
| Jiangle | 1.02 | 0.29 | 0.99 |
| Jianning | 1.02 | 0.19 | 0.99 |
| Shaxi | 0.97 | 0.26 | 0.99 |
| Taining | 0.99 | 0.21 | 0.99 |
| Shaowu | 1.03 | 0.31 | 0.99 |
| Shunchang | 1.03 | 0.32 | 0.99 |
| Jianou | 0.78 | 0.25 | 0.99 |
| Jianyang | 0.75 | 0.36 | 0.99 |
| Zhenghe | 1.01 | 0.14 | 0.99 |
| Songxi | 0.99 | 0.14 | 0.99 |
| Wuyishan | 0.82 | 0.27 | 1.00 |
| Pucheng | 1.05 | 0.26 | 0.99 |

Table S4 Temperature simulation results based on CMIP6bc

| Station | STD | RMSE | CC |
|---|---|---|---|
| Jiuxianshan | 1.10 | 0.66 | 0.96 |
| Gutian | 0.96 | 0.37 | 0.97 |
| Datian | 0.94 | 0.29 | 0.96 |
| Youxi | 0.95 | 0.30 | 0.96 |
| Dehua | 0.97 | 0.30 | 0.96 |
| Yongtai | 0.96 | 0.29 | 0.96 |
| Fuzhou | 0.96 | 0.32 | 0.96 |
| Changle | 0.88 | 0.28 | 0.96 |
| Minhou | 0.53 | 0.68 | 0.96 |
| Minqing | 0.94 | 0.27 | 0.96 |
| Sanming | 0.97 | 0.37 | 0.96 |
| Ninghua | 0.93 | 0.25 | 0.97 |
| Yongan | 0.95 | 0.30 | 0.96 |
| Shaxian | 0.96 | 0.33 | 0.97 |
| Qingliu | 0.91 | 0.26 | 0.97 |
| Liancheng | 0.93 | 0.32 | 0.96 |
| Guangze | 0.96 | 0.28 | 0.98 |
| Nanping | 0.98 | 0.37 | 0.97 |
| Jiangle | 0.95 | 0.27 | 0.97 |
| Jianning | 0.96 | 0.28 | 0.98 |
| Shaxi | 0.92 | 0.33 | 0.97 |
| Taining | 0.94 | 0.28 | 0.97 |
| Shaowu | 0.95 | 0.27 | 0.97 |
| Shunchang | 0.95 | 0.30 | 0.97 |
| Jianou | 0.79 | 0.32 | 0.97 |
| Jianyang | 0.73 | 0.41 | 0.97 |
| Zhenghe | 0.95 | 0.25 | 0.97 |
| Songxi | 0.94 | 0.25 | 0.97 |
| Wuyishan | 0.81 | 0.35 | 0.97 |
| Pucheng | 0.95 | 0.24 | 0.98 |

Table S5 Precipitation simulation results based on ERA5

| Station | STD | RMSE | CC |
|---------|-----|------|-----|
| Jiuxianshan | 1.20 | 0.71 | 0.80 |
| Gutian | 1.13 | 0.55 | 0.87 |
| Datian | 1.20 | 0.71 | 0.81 |
| Youxi | 1.35 | 0.72 | 0.86 |
| Dehua | 1.16 | 0.64 | 0.83 |
| Yongtai | 1.04 | 0.69 | 0.78 |
| Fuzhou | 0.94 | 0.75 | 0.74 |
| Changle | 0.76 | 0.89 | 0.62 |
| Minhou | 1.04 | 0.79 | 0.73 |
| Minqing | 1.10 | 0.65 | 0.81 |
| Sanming | 0.98 | 0.61 | 0.81 |
| Ninghua | 1.18 | 0.74 | 0.78 |
| Yongan | 1.12 | 0.61 | 0.84 |
| Shaxian | 0.97 | 0.56 | 0.84 |
| Qingliu | 1.16 | 0.72 | 0.79 |
| Liancheng | 1.36 | 0.89 | 0.75 |
| Guangze | 1.24 | 0.51 | 0.92 |
| Nanping | 0.97 | 0.53 | 0.85 |
| Jiangle | 1.07 | 0.45 | 0.91 |
| Jianning | 1.12 | 0.69 | 0.79 |
| Shaxi | 1.01 | 0.57 | 0.84 |
| Taining | 1.11 | 0.46 | 0.91 |
| Shaowu | 1.35 | 0.82 | 0.80 |
| Shunchang | 1.06 | 0.51 | 0.88 |
| Jianou | 0.96 | 0.54 | 0.85 |
| Jianyang | 0.99 | 0.46 | 0.89 |
| Zhenghe | 1.06 | 0.44 | 0.91 |
| Songxi | 1.17 | 0.55 | 0.89 |
| Wuyishan | 1.23 | 0.48 | 0.93 |
| Pucheng | 1.06 | 0.59 | 0.84 |

Table S6 Precipitation simulation results based on CMIP6bc

| Station | STD | RMSE | CC |
|---|---|---|---|
| Jiuxianshan | 1.19 | 0.98 | 0.61 |
| Gutian | 1.12 | 0.92 | 0.63 |
| Datian | 1.38 | 1.05 | 0.65 |
| Youxi | 1.11 | 0.85 | 0.68 |
| Dehua | 1.06 | 0.83 | 0.69 |
| Yongtai | 0.94 | 0.97 | 0.52 |
| Fuzhou | 0.94 | 0.94 | 0.57 |
| Changle | 0.82 | 0.70 | 0.73 |
| Minhou | 0.93 | 0.96 | 0.54 |
| Minqing | 1.16 | 0.97 | 0.60 |
| Sanming | 1.10 | 0.84 | 0.68 |
| Ninghua | 1.12 | 0.81 | 0.71 |
| Yongan | 1.26 | 1.04 | 0.59 |
| Shaxian | 0.97 | 0.82 | 0.66 |
| Qingliu | 1.23 | 0.98 | 0.63 |
| Liancheng | 1.31 | 1.11 | 0.56 |
| Guangze | 1.13 | 0.86 | 0.68 |
| Nanping | 0.98 | 0.75 | 0.71 |
| Jiangle | 1.13 | 0.84 | 0.69 |
| Jianning | 1.06 | 0.88 | 0.64 |
| Shaxi | 1.26 | 0.96 | 0.67 |
| Taining | 1.16 | 0.89 | 0.66 |
| Shaowu | 1.15 | 0.92 | 0.64 |
| Shunchang | 1.11 | 0.78 | 0.73 |
| Jianou | 1.01 | 0.80 | 0.68 |
| Jianyang | 1.13 | 0.89 | 0.65 |
| Zhenghe | 1.11 | 0.96 | 0.59 |
| Songxi | 1.15 | 0.90 | 0.66 |
| Wuyishan | 1.18 | 0.83 | 0.72 |
| Pucheng | 1.11 | 0.80 | 0.71 |

9.In the results and discussion, I could not find how the results show that considering the non-stationary characteristics leads to better or more reliable results. May you compare the results with previous studies in which the time series was assumed to be stationary?

Thank you for the constructive comment. We will enrich our results and add a discussion comparing them with previous studies in the revised manuscript.

Within the GAMLSS framework, we constructed both stationary and non-stationary models and evaluated their performance using the Akaike Information Criterion (AIC), adhering to the established principle that a smaller AIC value indicates a better model fit. The results show that, compared to the stationary model, the non-stationary model occupies more grid points (i.e., has a lower AIC value), indicating its stronger ability to adapt to the characteristics of the time series (Fig. 6). Furthermore, the comparative analysis of the return period results between the two models (Fig. 8) shows that the non-stationary model exhibits more extreme trends in compound extreme events (CEEs), reinforcing the importance of accounting for non-stationarity in such analyses. Additionally, the subsequent evaluation using the Filliben coefficient confirmed the goodness-of-fit of the selected models (Fig. S3), further validating our approach. We would like to clarify that the statement in the original manuscript claiming that the "non-stationary model is better than the stationary model" is misleading. A more accurate description is that the non-stationary model is more appropriate when accounting for temporal trends and mean-state changes, whereas the stationary model tends to underestimate the recurrence risk of CCEs based on the results.

Many previous studies have detected the non-stationary characteristics of hydro-meteorological variables (e.g., precipitation and runoff), indicating that their statistical properties are not constant over time (Shao et al., 2022; Awasthi et al., 2022; Slater et al., 2021). In a case study in Colombia (Gonzalez-Alvarez et al., 2018), the research compared return values under stationary and non-stationary conditions and found that rainfall estimates for the 10-year and 2-year return periods were significantly higher under non-stationary conditions, indicating that in extreme rainfall analysis, using a non-stationary model can better capture the increasing trend of rainfall than a stationary model. A low-flow frequency analysis study in Turkey indicated that non-stationary models outperform stationary models (Yılmaz and Muhammet, 2024), suggesting that when watershed hydrological relationships may change over time, relying solely on the stationary assumption could underestimate the associated risks. However, some studies have shown that in certain cases (De Luca and Galasso, 2018),

stationary models are sufficient, while non-stationary frameworks perform slightly better during periods with trends or variability. In this study, we assessed changes in the recurrence risk of CCEs based on the WRF-CMIP6-GAMLSS framework. The results indicate that CCEs in most areas of the MRB exhibit non-stationarity, primarily driven by mean-state shifts induced by climate warming. Overall, over time, stationary models systematically underestimate the risk of CCEs, particularly after 2045. These findings underscore the importance of incorporating non-stationary approaches in future climate risk assessments to improve the accuracy of extreme event predictions.

[Figure]

Fig. 6 Stationary and non-stationary characteristics for CCEs in the MRB (a-e and k-o), percentage 240 of non-stationary and stationary characteristics across five basins (f-j and p-t).

[Figure]

Fig. 8. Comparison of non-stationary (NS) and stationary (S) characteristics for CCEs under 20-, 50-, and 100- year return periods (a-j). Spatial distributions of trends in CCEs under 100-year return periods (k-t), 20- and 50-year return period result are provided in Supplement Fig. S4.

[Figure]

Fig. S3 Maps of the Filliben Coefficient for CCEs.

10.The English language also should be assessed more carefully.

Thank you for your suggestion. We will carefully revise the manuscript to further improve the clarity and quality of the English language.

**References**

Awasthi, C., Archfield, S.A., Ryberg, K.R., Kiang, J.E., Sankarasubramanian, A., 2022. Projecting Flood Frequency Curves Under Near-Term Climate Change. Water Resour. Res. 58, e2021WR031246. https://doi.org/10.1029/2021WR031246

Byun, U., Chang, E., Kim, J., Ahn, J., Cha, D., Min, S., Byun, Y., 2023. Investigation of Added Value in Regional Climate Models for East Asian Storm Track Analysis. J. Geophys. Res.: Atmos. 128, e2023JD039167. https://doi.org/10.1029/2023JD039167

De Luca, D.L., Galasso, L., 2018. Stationary and Non-Stationary Frameworks for Extreme Rainfall Time Series in Southern Italy. Water 10, 1477. https://doi.org/10.3390/w10101477

Gilbert, E., Pishniak, D., Torres, J.A., Orr, A., Maclennan, M., Wever, N., Verro, K., 2025. Extreme precipitation associated with atmospheric rivers over West Antarctic ice shelves: insights from kilometre-scale regional climate modelling. The Cryosphere 19, 597–618. https://doi.org/10.5194/tc-19-597-2025

Gonzalez-Alvarez, A., Coronado-Hernández, O.E., Fuertes-Miquel, V.S., Ramos, H.M., 2018. Effect of the Non-Stationarity of Rainfall Events on the Design of Hydraulic Structures for Runoff Management and Its Applications to a Case Study at Gordo Creek Watershed in Cartagena de Indias, Colombia. Fluids 3, 27. https://doi.org/10.3390/fluids3020027

Huang, Y., Xue, M., Hu, X., Martin, E., Novoa, H.M., McPherson, R.A., Liu, C., Chen, M.,

Hong, Y., Perez, A., Morales, I.Y., Ticona Jara, J.L., Flores Luna, A.J., 2024. Increasing frequency and precipitation intensity of convective storms in the Peruvian Central Andes: Projections from convection-permitting regional climate simulations. Quart. J. Royal Meteoro. Soc. 150, 4371–4390. https://doi.org/10.1002/qj.4820

Imran, H.M., Evans, J.P., 2025. Observational uncertainty in the added value of regional climate modelling over Australia. Clim. Dyn. 63, 73. https://doi.org/10.1007/s00382-024-07562-y

Jamal, K., Li, X., Chen, Y., Rizwan, M., Khan, M.A., Syed, Z., Mahmood, P., 2023. Bias correction and projection of temperature over the altitudes of the Upper Indus Basin under CMIP6 climate scenarios from 1985 to 2100. J. Water Clim. Change 14, 2490–2514. https://doi.org/10.2166/wcc.2023.180

Lin, C., Kjellström, E., Wilcke, R.A.I., Chen, D., 2022. Present and future European heat wave magnitudes: climatologies, trends, and their associated uncertainties in GCM-RCM model chains. Earth Syst. Dynam. 13, 1197–1214. https://doi.org/10.5194/esd-13-1197-2022

Shao, S., Zhang, H., Singh, V.P., Ding, H., Zhang, J., Wu, Y., 2022. Nonstationary analysis of hydrological drought index in a coupled human-water system: Application of the GAMLSS with meteorological and anthropogenic covariates in the Wuding River basin, China. J. Hydrol. 608, 127692. https://doi.org/10.1016/j.jhydrol.2022.127692

Singh, H., Najafi, M.R., Cannon, A.J., 2021. Characterizing non-stationary compound extreme events in a changing climate based on large-ensemble climate simulations. Clim. Dyn. 56, 1389–1405. https://doi.org/10.1007/s00382-020-05538-2

Slater, L., Villarini, G., Archfield, S., Faulkner, D., Lamb, R., Khouakhi, A., Yin, J., 2021. Global Changes in 20-Year, 50-Year, and 100-Year River Floods. Geophys. Res. Lett. 48, e2020GL091824. https://doi.org/10.1029/2020GL091824

Torrez-Rodriguez, L., Goubanova, K., Muñoz, C., Montecinos, A., 2023. Evaluation of temperature and precipitation from CORDEX-CORE South America and Eta-RCM regional climate simulations over the complex terrain of Subtropical Chile. Clim. Dyn. 61, 3195–3221. https://doi.org/10.1007/s00382-023-06730-w

Wu, L., Zheng, H., 2023. Regional Climate Effects of Irrigation under Central Asia Warming by 2.0 °C. Remote Sens. 15, 3672. https://doi.org/10.3390/rs15143672

Xu, Z., Han, Y., Tam, C.-Y., Yang, Z.-L., Fu, C., 2021. Bias-corrected CMIP6 global dataset for dynamical downscaling of the historical and future climate (1979–2100). Sci. Data 8,

293. https://doi.org/10.1038/s41597-021-01079-3

Yılmaz, M., Tosunoğlu, F., 2024. Non-stationary low flow frequency analysis under climate change. Theor. Appl. Climatol. 155, 7479–7497. https://doi.org/10.1007/s00704-024-05081-8

---

## Author Comment (AC2)

**RESPONSE TO REFEREES**

**Ms. Ref. No.:** Egusphere-2025-2438

**Title:** Non-Stationary Dynamics of Compound Climate Extremes: A WRF-CMIP6-GAMLSS Framework for Risk Reassessment in Southeastern China

**To Anonymous Referee #2**

We sincerely appreciate your thorough and constructive comments, which have been instrumental in improving our manuscript. We also apologize for the lack of clarity in the introduction and methodology sections of the manuscript, and we assure you that these issues will be addressed in the revised version. Please find the detailed point-by-point responses below.

**Major Comments:**

1.Throughout the manuscript (and also in the title), the authors state that this study addresses the 'risk' of CCEs. As risk assessment typically involves an (monetary) impact analysis (see e.g. UNDRR (2007) for the definition of natural hazard risk), it may be more accurate to frame the study in terms of frequency changes or recurrence, which the authors mentioned somewhere in the manuscript. Therefore, I recommend carefully reviewing and revising all uses of "risk" or "risk assessment" to avoid overstating the study's scope.

-Answer: We fully agree with and appreciate your suggestion regarding the use of the term "risk." As the reviewer pointed out, risk assessment typically involves an analysis of impacts, particularly economic or social impacts. From the perspective of disaster-inducing factors (hazard), we have not delved into analyzing the exposure of disaster-bearing bodies (such as society, ecosystems, etc.) and their capacity to withstand these impacts. Therefore, in this context, the term "risk" is indeed not entirely appropriate. Consequently, we plan to revise the title and related statements in the revised manuscript to more accurately reflect the actual content and scope of our study. Specifically, we will remove the phrase "Risk Reassessment" from the title, resulting in the revised title: Non-Stationary Dynamics of Compound Climate Extremes: A WRF-CMIP6-GAMLSS Framework for Southeastern China.

2.While research gaps are introduced in L60-62, they require clearer identification. There are a number of studies looking into the frequency changes in compound events, both in China and other parts of the world. To name a few: Zscheischler et al. (2018), Fang et al. (2025) npj climate and atmospheric science, Wu et al. (2023) Earth's Future, Ridder et al. (2022) npj climate and atmospheric science, and so on. How does the present study compare or advance

-Answer: Thank you for your valuable comment. We appreciate your suggestion and will will add a comparison with previous studies to highlight how the present study advances beyond these works and emphasizes its unique contributions in the revised manuscript.

Recent studies have increasingly focused on compound climate extremes (CCEs), highlighting their growing significance in the context of climate change. Zscheischler et al. (2018) were the first to clearly define the concept of compound events, emphasizing how the interaction of multiple climate and meteorological drivers can amplify extreme impacts. Building on this, Ridder et al. (2022) conducted the first global-scale assessment of the changes in compound events, specifically examining the co-occurrence of heatwaves and drought, extreme winds, and precipitation. Wu et al. (2023) revealed that under warming conditions, the risks associated with global compound pluvial–hot extreme events are projected to be significantly greater in the future than those observed during the historical period. Fang et al. (2025) investigated the future changes of sequential heatwaves and precipitation events (SHP) as well as concurrent drought and heatwave events (CDH) in China, with projections indicating an increase in both the frequency and intensity of these events.

*While large-scale studies play a crucial role in advancing our understanding of global climate change and extreme events, their practical relevance for disaster risk management and adaptation strategies in medium- and small-scale regions is relatively limited due to their lower spatial and temporal resolution.* To overcome this constraint, dynamical downscaling, which utilizes nested high-resolution regional climate models (RCMs), provides a critical technical pathway to investigate climate response mechanisms at fine-scales (Tapiador et al., 2020; Rahimi et al., 2024). In this context, over the past decade, an increasing number of studies have begun to use RCMs to obtain high-resolution climate information. Bozkurt et al. (2019) used the Regional Climate Model, version 4 (RegCM4), to evaluate the spatiotemporal variations of temperature and precipitation over the Pacific coast and the Andes Mountains. The results indicated that increasing the resolution effectively eliminates simulation errors caused by complex topography. McCrary et al. (2020) used multiple RCMs from the North American Coordinated Regional Downscaling Experiment (NA-CORDEX) to predict future snow changes in North America. They found that, particularly in high-elevation areas, the percentage of snow loss projected by GCMs was significantly higher than that projected by the RCMs. As an advanced convection-permitting RCM, the WRF model significantly enhances

the simulation capability for meteorological processes at 1-10 km scales through its fully compressible, non-hydrostatic dynamic core framework (Talbot et al., 2012). This high-resolution simulation capability gives the WRF model a unique advantage in capturing small-scale meteorological phenomena. Zhou et al. (2024) developed a 9 km resolution regional reanalysis dataset covering the Tibetan Plateau based on the WRF model, and demonstrated its superior applicability compared to the fifth generation European Centre for Medium-Range Weather Forecasts Reanalysis (ERA5). Yang et al. (2024) revealed that the WRF model provides better accuracy in simulating snow depth during the cold season in high-elevation regions compared to ERA5-Land.

Additionally, traditional extreme event analyses rely on stationarity assumptions, presuming that the probability and distributional parameters of climate variables are constant (Sun et al., 2018; Nerantzaki et al., 2023). However, driven by synergistic effects of global warming and anthropogenic forcing, extremes exhibit significant shifts in distributional characteristics (Gao et al., 2018). Therefore, traditional models are not suitable for evaluating extreme changes in the changing environment. To capture these changes, many studies have applied the Generalized Additive Models for Location, Scale, and Shape (GAMLSS) (Rigby and Stasinopoulos 2005) to address non-stationary problems in hydrological and meteorological extremes, enabling updated risk analysis of evolving climate extremes (Lei et al., 2021; Shao et al., 2022; Jin et al., 2023; Li et al., 2024). ***However, existing non-stationary analyses only focus on individual extremes, and the potential non-stationarity of CCEs has not been established. The comprehensive assessment of future changes in CCEs recurrence risk within a non-stationary framework is also lacking.***

***To address these research gaps, this study adopts a high-resolution approach, combining the WRF model with GAMLSS. This approach overcomes the limitations of traditional coarse-resolution models and addresses the shortcomings of stationary assumptions in analyzing compound climate extremes (CCEs).*** By focusing on the Minjiang River Basin (MRB), this research aims to explore four types of CCEs: hot-wet events (HW), hot-dry events (HD), cold-wet events (CW), and cold-dry events (CD). The analysis proceeds as follows (Fig. 1): Supplement Section S1 presents the validation of CMIP6bc applicability. Section 3.1 characterizes the spatio-temporal patterns of CCEs under both a middle-of-the-road scenario (SSP2-4.5) and a high-emissions scenario (SSP5-8.5). The non-stationarity detection of CCEs is described in Section 3.2. The recurrence risk changes in CCEs under non-stationary

conditions is evaluated in Section 3.3. The work establishes a scientific basis for addressing the environmental and climatic challenges posed by CCEs, thereby contributing to effective strategies for regional sustainability and climate resilience.

**References**

Bozkurt, D., Rojas, M., Boisier, J.P., Rondanelli, R., Garreaud, R., Gallardo, L., 2019. Dynamical downscaling over the complex terrain of southwest South America: present climate conditions and added value analysis. Clim. Dyn. 53, 6745–6767. https://doi.org/10.1007/s00382-019-04959-y

Fang, P., Wang, T., Yang, D., Tang, L., Yang, Y., 2025. Substantial increases in compound climate extremes and associated socio-economic exposure across China under future climate change. npj Clim. Atmos. Sci. 8, 17. https://doi.org/10.1038/s41612-025-00910-7

Gao, L., Huang, J., Chen, X., Chen, Y., Liu, M., 2018. Contributions of natural climate changes and human activities to the trend of extreme precipitation. Atmos. Res. 205, 60–69. https://doi.org/10.1016/j.atmosres.2018.02.006

Jin, H., Willems, P., Chen, X., Liu, M., 2023. Nonstationary flood and its influencing factors analysis in the Hanjiang River Basin, China. J. Hydrol. 625, 129994. https://doi.org/10.1016/j.jhydrol.2023.129994

Lei, X., Gao, L., Ma, M., Wei, J., Xu, L., Wang, L., Lin, H., 2021. Does non-stationarity of extreme precipitation exist in the Poyang Lake Basin of China? J. Hydrol.: Reg. Stud. 37, 100920. https://doi.org/10.1016/j.ejrh.2021.100920

Li, M., Feng, Z., Zhang, M., Yao, Y., 2024. Influence of large-scale climate indices and regional meteorological elements on drought characteristics in the Luanhe River Basin. Atmos. Res. 300, 107219. https://doi.org/10.1016/j.atmosres.2024.107219

McCrary, R.R., Mearns, L.O., Abel, M.R., Biner, S., Bukovsky, M.S., 2022. Projections of North American snow from NA-CORDEX and their uncertainties, with a focus on model resolution. Climatic Change 170, 20. https://doi.org/10.1007/s10584-021-03294-8

Nerantzaki, S.D., Papalexiou, S.M., Rajulapati, C.R., Clark, M.P., 2023. Nonstationarity in High and Low-Temperature Extremes: Insights From a Global Observational Data Set by Merging Extreme-Value Methods. Earth's Future 11, e2023EF003506. https://doi.org/10.1029/2023EF003506

Rahimi, S., Huang, L., Norris, J., Hall, A., Goldenson, N., Risser, M., Feldman, D.R., Lebo,

Z.J., Dennis, E., Thackeray, C., 2024. Understanding the Cascade: Removing GCM Biases Improves Dynamically Downscaled Climate Projections. Geophys. Res. Lett. 51, e2023GL106264. https://doi.org/10.1029/2023GL106264

Ridder, N.N., Ukkola, A.M., Pitman, A.J., Perkins-Kirkpatrick, S.E., 2022. Increased occurrence of high impact compound events under climate change. npj Clim. Atmos. Sci. 5, 3. https://doi.org/10.1038/s41612-021-00224-4

Rigby, R.A., Stasinopoulos, D.M., 2005. Generalized Additive Models for Location, Scale and Shape. J. R. Stat. Soc. Series C: Appl. Stat. 54, 507–554. https://doi.org/10.1111/j.1467-9876.2005.00510.x

Shao, S., Zhang, H., Singh, V.P., Ding, H., Zhang, J., Wu, Y., 2022. Nonstationary analysis of hydrological drought index in a coupled human-water system: Application of the GAMLSS with meteorological and anthropogenic covariates in the Wuding River basin, China. J. Hydrol. 608, 127692. https://doi.org/10.1016/j.jhydrol.2022.127692

Sun, F., Roderick, M.L., Farquhar, G.D., 2018. Rainfall statistics, stationarity, and climate change. Proc. Natl. Acad. Sci. U.S.A. 115, 2305–2310. https://doi.org/10.1073/pnas.1705349115

Talbot, C., Bou-Zeid, E., Smith, J., 2012. Nested Mesoscale Large-Eddy Simulations with WRF: Performance in Real Test Cases. J. Hydrometeorol. 13, 1421–1441. https://doi.org/10.1175/JHM-D-11-048.1

Tapiador, F.J., Navarro, A., Moreno, R., Sánchez, J.L., García-Ortega, E., 2020. Regional climate models: 30 years of dynamical downscaling. Atmos. Res. 235, 104785. https://doi.org/10.1016/j.atmosres.2019.104785

Wu, H., Su, X., Singh, V.P., 2023. Increasing Risks of Future Compound Climate Extremes With Warming Over Global Land Masses. Earth's Future 11, e2022EF003466. https://doi.org/10.1029/2022EF003466

Yang, T., Chen, X., Hamdi, R., Li, Q., Cui, F., Li, L., Liu, Y., De Maeyer, P., Duan, W., 2024. Assessment of snow simulation using Noah-MP land surface model forced by various precipitation sources in the Central Tianshan Mountains, Central Asia. Atmos. Res. 300, 107251. https://doi.org/10.1016/j.atmosres.2024.107251

Zhou, P., Tang, J., Ma, M., Ji, D., Shi, J., 2024. High resolution Tibetan Plateau regional reanalysis 1961-present. Sci. Data 11, 444. https://doi.org/10.1038/s41597-024-03282-4

Zscheischler, J., Westra, S., Van Den Hurk, B.J.J.M., Seneviratne, S.I., Ward, P.J., Pitman, A.,

AghaKouchak, A., Bresch, D.N., Leonard, M., Wahl, T., Zhang, X., 2018. Future climate risk from compound events. Nature Clim. Change 8, 469–477. https://doi.org/10.1038/s41558-018-0156-3

3.Fig. 1 outlines the key methodological steps in this study. Many unintroduced acronyms e.g. WRFOUT, CC and PBIAS are used without prior explanation, which may reduce the readability and understanding of the framework.

-Answer: Thank you for your comment. We agree with your suggestion and will revise Figure 1 in the revised manuscript to ensure better readability for the readers.

[Figure]

Fig. 1. Flowchart of CCEs projection in a non-stationary framework.

4.Please elaborate on the types of CCEs commonly occurring during the flood season (Line 99). Including examples of historical events would help illustrate this.

-Answer: Thank you for your constructive comment. We agree that providing more details in the revised manuscript.

The basin displays spatio-temporal heterogeneity in precipitation, with flood seasons from April to September that often accompany CCEs. Particularly in the late flood season (July to

September), the MRB experiences frequent typhoon-related compound disasters: the upper and middle reaches are commonly affected by typhoon-rainstorm-landslide events, while the lower reaches face high occurrences of typhoon-rainstorm-urban waterlogging and typhoon-rainstorm-flood events (Yang et al., 2025). In 2023, for example, Typhoon Doksuri (No. 2305) caused approximately 66,794 people to be affected in Fuzhou, the downstream city of the MRB, with direct economic losses reaching 588 million RMB (Yan et al., 2024). In addition, the region also exhibits a climate characteristic of concurrent rainfall and heat, with CCEs frequently occurring during the warm season, driven by high temperatures and heavy rainfall (Sun et al., 2025).


ii. Why would the ERA5 serve as the benchmark not the observed data (L113-115)?

-Answer: Thank you for your question. We would like to clarify that, in fact, we have three sets of data: observed data, WRF-ERA5, and WRF-CMIP6bc. ERA5, as a widely used WRF-driven dataset, is utilized here as a reference for the simulation results. Indeed, the actual validation data comes from the meteorological station observations (Figure S1).

[Figure]

Figure S1. Evaluation of WRF-simulated precipitation and temperature over the MRB (2005-2014). Spatial patterns of precipitation (a–c), temperature (d–f). Temporal evolution of precipitation (g) and temperature (h). Panels (i–l) present sub-basin comparisons of precipitation and temperature from ERA5 (i, k) and CMIP6bc (j, l).

iii. How were the spatial results constructed from the observed data at 30 stations?

-Answer: Thank you for your question. The spatial distribution results of the CCEs were constructed by performing spline interpolation on the observed data from the 30 stations in the MRB using Arc Geographic Information System (ArcGIS). To eliminate the interpolation errors, we extracted the nearest WRF grid points for each meteorological station and compared

them by plotting a Taylor diagram. We believe that using both spatial interpolation and point-to-point validation can effectively reduce uncertainty in the results. We will provide a more detailed explanation of this approach in the revised manuscript.

iv. Please define in the manuscript what CC and PBIAS mean, and clarify whether they represent averages across all grid cells.

-Answer: Thank you for your comment. We will add definitions of the evaluation metrics in the revised manuscript (Table 2). We would also like to clarify that in the spatial maps, CC and PBIAS represent the averages across all stations, whereas in the Taylor diagrams, CC, RMSE, and STD are calculated based on the nearest WRF grid point to each station.

Table 2 Definition of evaluation criteria.

| Metric | Formula | Optimal value | Range |
|--------|---------|---------------|-------|
| CC | $CC = \dfrac{\sum_{i=1}^{n}(O_i - \overline{O_i})(M_i - \overline{M_i})}{\sqrt{\sum_{i=1}^{n}(O_i - \overline{O_i})^2}\sqrt{\sum_{i=1}^{n}(M_i - \overline{M_i})^2}}$ | 1 | (0, 1) |
| PBIAS | $PBIAS = \sum_{i=1}^{n} \dfrac{M_i - O_i}{O_i}$ | 0 | (-∞, +∞) |
| RMSE | $RMSE = \sqrt{\dfrac{1}{n}\sum_{i=1}^{n}(M_i - O_i)}$ | 0 | (0, +∞) |
| STD | $STD = \sqrt{\dfrac{1}{n}\sum_{i=1}^{n}(O_i - \bar{O})^2}$ | 0 | (0, +∞) |

Notes: Where $O_i$ presents hydrometeorological data at the $i$ station, $M_i$ presents data at the WRF grid point closest to the $i$ station, $n$ is the numbers of stations.

v. What are subplots a-f in Figure S1 based on? The mean value per grid over the 10-year historic period? Please add necessary details on what you are comparing.

Thank you for your question. Subplots a–f in Figure S1 represent the interpolated annual mean values at each meteorological station and the nearest WRF grid point over the 10-year historical period. We will add this description in the revised manuscript.

vi. Is there also an explanation on those misestimated CMIP6bc temperatures? And why are the results particularly worse in the downstream sub-basin of the MRB?

-Answer: Thank you for your question. We will provide an explanation for this issue in the supplement.

The WRF model's simulation of temperature in complex terrain is mainly influenced by factors such as radiation transfer, surface type, and meteorological initial conditions (Jiménez-Esteve et al., 2018; Liu et al., 2019; Lu et al., 2021). High-altitude areas typically experience

stronger radiation effects, especially in mountainous regions with thinner atmospheres. The WRF model may have errors in simulating radiation transfer, leading to temperatures in high-altitude areas being lower than actual conditions (Varga et al., 2020). In addition, the downstream area of the MRB is an urban agglomeration, where urban areas typically experience stronger radiation and heat accumulation effects. These effects may not be fully accounted for in the model, leading to simulated temperatures being higher than actual conditions (Li et al., 2014; Chen et al., 2025). Ntoumos et al. (2023) also revealed that the WRF model tends to overestimate the maximum temperatures and underestimate the minimum temperatures, with the errors being closely related to the geographic location.


ii.Did you consider the spatially compound events, i.e. the same CCE event occur at multiple locations simultaneously? If not, please discuss if the proposed approach may double count CCEs and how does this affect your conclusion.

-Answer: Thank you for your question. You've raised a very important point, and we fully understand your concern about whether the simultaneous occurrence of the same compound climate extreme event (CCE) across multiple grid points might lead to double counting, potentially affecting the reliability of the results. In response, we would like to clarify that when calculating the compound extreme indices, we perform the calculations independently for each grid point. For example, in Fig. 3, subplots a-j show the annual mean spatial distribution of each type of CCE for each grid point over the 40-year period, rather than the total sum. Therefore, even if the same event occurs at multiple grid points simultaneously, this will not

lead to double counting (For example, even if the same CCE occurs on the same day at two grid points, it will still be counted as one day after averaging.). Similarly, for subplots k-o, the time series variations are averaged across the grid points, representing the mean values of the entire basin, not the total sum, so there is no issue of double counting in these results either.

Nevertheless, we appreciate the insightful nature of your comment. Spatially compound events are a more complex concept, and we intend to conduct separate analyses for such cases in future studies.

c.Please include the results of your sensitivity experiments (L130) in the supplementary materials.

-Answer: Thank you for your suggestion. We will include the results of the sensitivity experiments in the supplementary materials.

In our previous study, we conducted a detailed sensitivity analysis and optimization of parameterization schemes specifically for the MRB (Lin et al., 2023). We focused on two schemes that have the most significant impact on precipitation: microphysics and cumulus convection. By cross-combining these schemes, we developed 16 combinations (as shown in Table S2) and assessed their performance in simulating precipitation across different magnitudes. Through a comprehensive evaluation of the temporal (Fig. S1 and Fig. S3) and spatial (Fig. S3 and Table S3) characteristics, we determined that the 9th configuration (Lin and NT) is the most suitable for simulating extreme precipitation events in the MRB. As a result, this configuration was adopted in the present study to ensure the most accurate simulation of extreme precipitation events in this region.

Table S2 Parameterization scheme combinations design.

| Combinations | Microphysics scheme (MP) | Cumulus scheme (CU) |
| --- | --- | --- |
| EXP1 | WSM6 | Betts-Miller-Janjic (BMJ) |
| EXP2 | WSM6 | Betts-Miller-Janjic (BMJ) |
| EXP3 | WSM6 | Betts-Miller-Janjic (BMJ) |
| EXP4 | WSM6 | Betts-Miller-Janjic (BMJ) |
| EXP5 | WDM6 | Kain-Fritsch (KF) |
| EXP6 | WDM6 | Kain-Fritsch (KF) |
| EXP7 | WDM6 | Kain-Fritsch (KF) |
| EXP8 | WDM6 | Kain-Fritsch (KF) |
| EXP9 | Purdue Lin (Lin) | New Tiedtke (NT) |
| EXP10 | Purdue Lin (Lin) | New Tiedtke (NT) |

| | | |
|---|---|---|
| EXP11 | Purdue Lin (Lin) | New Tiedtke (NT) |
| EXP12 | Purdue Lin (Lin) | New Tiedtke (NT) |
| EXP13 | Thompson | Grell-Devenyi (GD) |
| EXP14 | Thompson | Grell-Devenyi (GD) |
| EXP15 | Thompson | Grell-Devenyi (GD) |
| EXP16 | Thompson | Grell-Devenyi (GD) |

[Figure]

Fig. S2. Box plot of TS scores for 24-hour accumulated precipitation simulated by WRF.

Table S3 Evaluation metrics for total accumulated precipitation

| Combination | TS (light) | TS (moderate) | TS (heavy) | TS (torrential) | $\overline{TS}$ | $\overline{POD}$ | $\overline{FAR}$ |
|---|---|---|---|---|---|---|---|
| EXP1 | 0.22 | 0.11 | 0.07 | 0.15 | 0.14 | 0.24 | 0.71 |
| EXP2 | 0.22 | 0.10 | 0.14 | 0.08 | 0.14 | 0.24 | 0.69 |
| EXP3 | 0.24 | 0.11 | 0.09 | 0.16 | 0.15 | 0.27 | 0.66 |
| EXP4 | 0.22 | 0.05 | 0.14 | 0.14 | 0.14 | 0.25 | 0.70 |
| EXP5 | 0.14 | 0.07 | 0.07 | 0.11 | 0.10 | 0.16 | 0.71 |
| EXP6 | 0.12 | 0.09 | 0.09 | 0.07 | 0.09 | 0.15 | 0.74 |
| EXP7 | 0.18 | 0.09 | 0.10 | 0.12 | 0.12 | 0.20 | 0.64 |
| EXP8 | 0.16 | 0.09 | 0.11 | 0.14 | 0.12 | 0.19 | 0.68 |
| EXP9 | 0.26 | 0.11 | 0.12 | 0.16 | 0.16 | 0.29 | 0.66 |
| EXP10 | 0.22 | 0.10 | 0.11 | 0.11 | 0.14 | 0.24 | 0.74 |
| EXP11 | 0.24 | 0.12 | 0.13 | 0.14 | 0.16 | 0.23 | 0.55 |

| | | | | | | | |
|---|---|---|---|---|---|---|---|
| EXP12 | 0.25 | 0.10 | 0.10 | 0.13 | 0.15 | 0.24 | 0.71 |
| EXP13 | 0.22 | 0.10 | 0.07 | 0.11 | 0.12 | 0.21 | 0.67 |
| EXP14 | 0.21 | 0.08 | 0.08 | 0.06 | 0.11 | 0.20 | 0.74 |
| EXP15 | 0.24 | 0.09 | 0.10 | 0.13 | 0.14 | 0.25 | 0.70 |
| EXP16 | 0.24 | 0.12 | 0.13 | 0.12 | 0.15 | 0.26 | 0.67 |

Note: TS (light rain, moderate rain, heavy rain, torrential rain) refers to the average of daily 24-hour accumulated precipitation. $\overline{TS}$、$\overline{POD}$、$\overline{FAR}$ represent the average values of the scores for the four precipitation levels.

[Figure]

Fig. S3. Spatial distribution of biases of total accumulated precipitation from WRF parametrization scheme sensitivity experiments.

**References**

Lin, S., Zhang, Y., Sun, S., Guan, X., Jiang, C., Gao, L., 2023. Sensitivity study of WRF parameterization schemes and initial fields on simulation of rainstorm in the Minjiang River basin. Pearl River 44(10): 35-46+61. https://doi: 10.3969/j.issn.1001-9235.2023.10.004 (in Chinese)

d.While Sect. 2.4.2 introduces the GAMLSS model, it lacks the detail on how WRF results are used in this model. The authors may include suitable examples when explaining the proposed approach. For example, how are the daily temperature and precipitation or the identified CCEs used in the model?

-Answer: Thank you for your suggestion. In our study, the output from the WRF model (precipitation and temperature) is used as meteorological variables to identify and calculate compound climate extreme events (CCEs), including hot-wet events (HW), hot-dry events (HD), cold-wet events (CW), and cold-dry events (CD). After calculating the CCEs for each grid point, these events are input into the GAMLSS model for further analysis. In the GAMLSS model, time (year) is used as the independent variable (x), and the number of days per year for each type of CCE is treated as the dependent variable (y), thereby enabling the calculation of the non-stationary characteristics of each CCE.

e.L158-159: "we fit non-stationary GAMLSS models with two parameters (mean, variance) and four parameters (mean, variance, skewness, kurtosis) at each grid point, selecting the optimal model for subsequent analysis." What is the difference between the mean and variance in the first mention (two parameters) and in the following (four parameters)? And how are the changes between the current and future CCE frequency estimated and in what unit/form?

-Answer: We appreciate the reviewer's attention to this point. We employed two types of GAMLSS models to capture potential changes in the distribution of meteorological variables. The first is the traditional two-parameter location–scale model (mean $\mu$ and variance $\sigma$), which assumes a fixed distributional shape. The second is a more flexible four-parameter location–scale–shape model (mean $\mu$, variance $\sigma$, skewness $\nu$, and kurtosis $\tau$), which allows the distributional shape to vary over time. These extended distributions retain the mean and variance parameters but include two extra shape parameters to capture asymmetry and tail behavior. We evaluate these models by comparing their goodness of fit and then select the model that best represents the data distribution for subsequent analyses. Additionally, skewness and kurtosis are unitless statistics that measure the asymmetry and tail thickness of the data

distribution, respectively. The unit of frequency change is days per decade, and it was calculated by using linear regression based on annual data to estimate the trend of frequency change.

6.Please describe in the methodology how metrics such as annual event distribution, seasonal variations, and non-stationary characteristics were calculated.

-Answer: Thank you for your suggestion. We will include this additional information in the methodology section. In calculating the interannual and seasonal variations of CCEs, we used different threshold values. Specifically, for interannual variation, we sorted the precipitation and temperature data over a 40-year period and determined the thresholds based on the 10th and 90th percentiles to identify CCEs. For seasonal variations, we separately extracted the precipitation and temperature data for the summer (JJA) and winter (DJF) seasons, applying the same sorting method to calculate the respective thresholds, thus analyzing the distribution characteristics of CCEs for each season.

Regarding the calculation of stationarity, we employed the GAMLSS to fit the changes in the mean and variance of CCEs. To assess their stationarity over time, we consider the CCEs to be stationary if both the mean and variance remain stable. If either the mean or variance shows significant variation, the CCEs are considered non-stationary.

7.In Fig. 3 (a-j), do the annual spatial values refer to the mean yearly event number over 2025-2065? What does d/10a mean?

-Answer: Thank you for your question. We apologize for the previous lack of clarity in our description. In Fig. 3 (a-j), the annual spatial values represent the mean yearly event number over the period 2025-2065. Regarding the term "d/10a," we will revise it to "days per decade" to ensure clearer understanding.

8.When looking at the temporal changes in CCEs (Fig. 3 k-o), it is interesting to see the projections for two SSPs sometimes show a completely different year-to-year trend. For example, for hot-dry events, the value in blue increases from 2032 to 2033 while the red one decreases. There are also similar discrepancies for different years and different compound events. Is there an explanation for this?

-Answer: Thank you for pointing out this important issue. The inconsistency, or even opposite trends, of extreme events in some regions under the SSP2-4.5 and SSP5-8.5 pathways in climate simulations is reasonable and expected. This is mainly due to the dominant role of internal climate variability at the regional scale. Although the greenhouse gas forcing in SSP5-

8.5 is stronger, the global warming and atmospheric circulation responses (such as changes in jet stream positions or shifts in storm tracks) caused by it differ spatially from those in SSP2-4.5. These differences interact in complex ways with decadal-scale internal oscillations, such as the Pacific Decadal Oscillation (PDO) or the Atlantic Multidecadal Oscillation (AMO). During specific periods and in certain regions, these strong natural variability signals may temporarily mask or even reverse the long-term trends driven by external forcing, leading to different short-term evolution paths under the two scenarios. Additionally, similar cases have been observed in previous studies (Fig. 5 (Wu et al., 2023); Fig. 7 (Ren et al., 2023); Fig.2 (Fang et al., 2025)), demonstrating that short-term variability under different emission scenarios is quite common.


**4.3 Frequency of recurrence systematically underestimated by stationary models**

Our comparison between stationary and non-stationary models reveals that the latter captures a significant increase in recurrence risks, particularly for 100-year CCEs (3.12 days per decade under SSP5-8.5). Stationary models systematically underestimate these risks after 2045, consistent with global studies showing that conventional extreme value models fail to capture escalating severities under climate change (Feng et al., 2020; Xu et al., 2025). The stronger non-stationary response of 100-year events highlights the heightened vulnerability of high-impact, low-probability extremes—a critical insight for infrastructure design and disaster preparedness. The west-to-east gradient in recurrence risk, with hotspots in the Shaxi River Basin, may be attributed to topographic and land-surface heterogeneity, which modulate local hydroclimatic responses (Zheng et al., 2023; Zhang et al., 2025).

**4.4 Methodological advances and limitations**

Our integrated "bias-corrected CMIP6–WRF dynamical downscaling–GAMLSS" framework represents a significant methodological advancement over the direct use of raw GCM outputs or purely statistical downscaling for projecting CCEs. By resolving mesoscale circulations and explicitly simulating convective processes, our approach more faithfully captures the fine-scale spatiotemporal heterogeneity of precipitation and temperature fields in complex terrain, a capability that statistical methods, reliant on historically derived statistical relationships, fundamentally lack (Gutmann et al., 2012; Rahimi et al., 2024). Crucially, initializing the WRF model with a bias-corrected CMIP6 dataset mitigates the propagation and amplification of inherent GCM systematic errors, a strategy proven to enhance the credibility of regional climate projections (Zhang et al., 2024; Rahimi et al., 2024). Nonetheless, certain limitations persist. Even at convection-permitting resolution (3 km), the WRF model exhibits systematic biases in simulating orographic precipitation, a well-documented challenge often stemming from uncertainties in microphysical parameterization schemes and the representation of land-atmosphere energy and moisture exchanges over mountainous regions (Talbot et al., 2012; Zhang et al., 2025). Furthermore, while statistically robust, our current non-stationary GAMLSS framework employs time merely as a proxy covariate for climate change. This approach effectively detects and projects temporal trends in risk but falls short of elucidating the underlying physical drivers, such as the specific roles of evolving large-scale circulation patterns or soil moisture-atmosphere feedbacks. To overcome these constraints and solidify the

physical foundations of our projections, future work should focus on three promising avenues: first, explicitly embedding physical drivers like atmospheric circulation indices, antecedent soil moisture, or global mean temperature as covariates within the GAMLSS to establish a clearer causal chain from forcing to statistical response (Zeng et al., 2024; Ma et al., 2025); second, leveraging machine learning, such as convolutional neural networks, for the statistical post-processing of WRF outputs to correct systematic biases, or developing hybrid physics-informed machine learning models as a complementary approach to dynamical downscaling (Yin et al., 2021; Xie et al., 2023); and third, systematically quantifying the full cascade of uncertainty from GCMs through downscaling to statistical modeling, ideally through a super-ensemble of multiple CMIP6 models and WRF physical parameterizations, to provide probabilistic risk estimates crucial for informed decision-making.


Table S1 CMIP6 models used in CMIP6bc.

| No. | Model | Institution | Approximate grid spacing |
|---|---|---|---|
| 1 | ACCESS-CM2 | Commonwealth Scientific and Industrial Research Organisation (Australia) | $1.875° \times 1.25°$ |
| 2 | ACCESS-ESM1–5 | Commonwealth Scientific and Industrial Research Organisation (Australia) | $1.875° \times 1.25°$ |
| 3 | CanESM5 | Canadian Centre for Climate Modelling and Analysis (Canada) | $2.81° \times 2.81°$ |
| 4 | BCC-CSM2-MR | Beijing Climate Center (China) | $1.125° \times 1.125°$ |
| 5 | FGOALS-f3-L | Institute of Atmospheric Physics, Chinese Academy of Sciences (China) | $1.25° \times 1°$ |
| 6 | FGOALS-g3 | Institute of Atmospheric Physics, Chinese Academy of Sciences (China) | $2° \times 2.25°$ |
| 7 | EC-Earth3 | European EC-Earth Consortium (Europe) | $0.70° \times 0.70°$ |
| 8 | EC-Earth3-Veg | European EC-Earth Consortium (Europe) | $0.70° \times 0.70°$ |
| 9 | IPSL-CM6A-LR | Institute Pierre Simon Laplace (France) | $2.5° \times 1.26°$ |
| 10 | AWI-CM-1-1-MR | Alfred Wegener Institute, Helmholtz Centre for Polar and Marine Research (Germany) | $0.94° \times 0.94°$ |
| 11 | MPI-ESM1-2-HR | Max Planck Institute for Meteorology (Germany) | $0.94° \times 0.94°$ |
| 12 | MPI-ESM1-2-LR | Max Planck Institute for Meteorology (Germany) | $1.875° \times 1.875°$ |
| 13 | MIROC6 | Japan Agency for Marine-Earth Science and Technology (Japan) | $1.41° \times 1.41°$ |
| 14 | MRI-ESM2-0 | Meteorological Research Institute, Japan Meteorological Agency (Japan) | $1.125° \times 1.125°$ |
| 15 | NorESM2-LM | Norwegian Climate Center (Norway) | $2.5° \times 1.875°$ |
| 16 | CESM2 | Climate and Global Dynamics Laboratory, National Center for Atmospheric Research (USA) | $1.25° \times 0.94°$ |
| 17 | CESM2-WACCM | Climate and Global Dynamics Laboratory, National Center for Atmospheric Research (USA) | $1.25° \times 0.94°$ |
| 18 | GFDL-ESM4 | Geophysical Fluid Dynamics Laboratory, National Oceanic and Atmosphere Administration (USA) | $1.25° \times 1.0°$ |


-Answer: Thank you for your suggestion. We will modify the manuscript accordingly and use a different color for the meteorological stations in the revised version to avoid confusion with the elevation data (Fig. 2).

[Figure]

Fig. 2. Study area and model configuration. (a) Topographic features of the MRB (m) and (b) Model configuration with 9-km (D01) and 3-km (D02) nested domains (Zhang et al., 2025). Basemap source: © Esri, https://services.arcgisonline.com

-Answer: We sincerely appreciate your valuable comments, and we apologize for the inconsistency in referring to the figures. Following your suggestion, we will ensure format consistency in the revised manuscript to align with the figure captions. Thank you once again for your careful review and valuable suggestions.